# Predictive Value of *SLCO1B1* c.521T>C Polymorphism on Observed Changes in the Treatment of 1136 Statin-Users

**DOI:** 10.3390/genes14020456

**Published:** 2023-02-10

**Authors:** Marleen E. Jansen, Tessel Rigter, Thom M. C. Fleur, Patrick C. Souverein, W. M. Monique Verschuren, Susanne J. Vijverberg, Jesse J. Swen, Wendy Rodenburg, Martina C. Cornel

**Affiliations:** 1Department of Clinical Genetics, Amsterdam Public Health Research Institute, Personalized Medicine, Amsterdam UMC, Vrije Universiteit Amsterdam, Section Community Genetics, 1081 HV Amsterdam, The Netherlands; 2Centre for Health Protection, National Institute for Public Health and the Environment, 3721 MA Bilthoven, The Netherlands; 3Division of Pharmacoepidemiology & Clinical Pharmacology, Utrecht Institute for Pharmaceutical Sciences, Utrecht University, 3508 TB Utrecht, The Netherlands; 4Centre for Nutrition, Prevention and Health Services, National Institute for Public Health and the Environment, 3721 MA Bilthoven, The Netherlands; 5Julius Center for Health Sciences and Primary Care, University Medical Center Utrecht, Utrecht University, 3508 GA Utrecht, The Netherlands; 6Department of Pulmonary Medicine and Amsterdam Public Health Research Institute, Personalized Medicine, Amsterdam UMC, University of Amsterdam, 1105 AZ Amsterdam, The Netherlands; 7Department of Clinical Pharmacy & Toxicology, Leiden University Medical Centre, 2300 RC Leiden, The Netherlands

**Keywords:** statins, pharmacogenomics, primary care, screening, adverse drug reactions

## Abstract

Pharmacogenomic testing is a method to prevent adverse drug reactions. Pharmacogenomics could be relevant to optimize statin treatment, by identifying patients at high risk for adverse drug reactions. We aim to investigate the clinical validity and utility of pre-emptive pharmacogenomics screening in primary care, with *SLCO1B1* c.521T>C as a risk factor for statin-induced adverse drug reactions. The focus was on changes in therapy as a proxy for adverse drug reactions observed in statin-users in a population-based Dutch cohort. In total, 1136 statin users were retrospectively genotyped for the *SLCO1B1* c.521T>C polymorphism (rs4149056) and information on their statin dispensing was evaluated as cross-sectional research. Approximately half of the included participants discontinued or switched their statin treatment within three years. In our analyses, we could not confirm an association between the *SLCO1B1* c.521T>C genotype and any change in statin therapy or arriving at a stable dose sooner in primary care. To be able to evaluate the predictive values of *SLCO1B1* c.521T>C genotype on adverse drug reactions from statins, prospective data collection of actual adverse drug reactions and reasons to change statin treatment should be facilitated.

## 1. Introduction

Pharmacogenomic testing (PGx) is a method to prevent adverse drug reactions (ADRs). PGx could be relevant to optimize statin treatment [1,2]. Statins reduce low-density lipoprotein cholesterol (LDL-c) levels and lower the risk for cardiovascular events [3]. Statins are prescribed to millions of patients and are usually well-tolerated [4]; however, statin-related myopathy (SRM) is estimated to manifest in 10–25% of patients in clinical practice [5,6,7,8,9]. Approximately 30% of patients discontinue treatment without consulting their physician within their first year, most likely due to ADRs such as mild SRM [5,6,9]. Patients who do not consult their physician might be at increased risk for cardiovascular events, since they do not get alternative treatment, with inherent medical and economic consequences [10,11]. By preemptive screening PGx related to statins, SRM and subsequent non-compliance can be reduced and patients can receive a more effective treatment, resulting in better health outcomes. Therefore, we studied the influence of one PGx-variant on SRM.

SRM ranges from mild myalgia (skeletal muscle pain without evidence of muscle degradation) to rhabdomyolysis (severe skeletal muscle damage with acute kidney injury) [12]. The risk of developing SRM depends on several factors. The risk is higher for more lipophilic statins, such as simvastatin, especially at a high dose [5,13]. Secondly, demographics and lifestyle factors contribute to this risk, such as higher age, lower body mass index (BMI), and being female [5,6,9]. Furthermore, the solute carrier organic anion transporter family member 1B1 (*SLCO1B1*) c.521T>C polymorphism (rs4149056) is associated with SRM [5,9,14,15,16,17]. To prevent SRM, the Clinical Pharmacogenetics Implementation Consortium (CPIC) and the Dutch Pharmacogenetics Working Group (DPWG) provide PGx guidelines for dosing simvastatin and atorvastatin based on the *SLCO1B1* c.521T>C SNP [12,18,19,20]. In the CPIC guideline on *SLCO1B1* and SRM, it is stated that, theoretically, 30 patients need to be genotyped to prevent one ADR.

In a meta-analysis, Xiang et al. (2018) concluded that the association between SLCO1B1 and SRM is not consistent [21]. While the overall odds ratios seem to indicate there is an association with SRM, especially for simvastatin, significant results were not confirmed in the included fourteen studies. The genome-wide study of the SEARCH Collaborative Group in 2008 was the first to report a strong association between the single nucleotide polymorphism (SNP) c.521T>C (rs4149056) in *SLCO1B1* and SRM in patients treated with 80 mg simvastatin daily [22]. The risk for myopathy in CC homozygotes was higher than in TT homozygotes (odds ratio (OR) 16.9; 95% CI 4.7–61.1; 85 cases and 90 controls). Link et al. (2008) replicated this result in participants of the Heart Protection Study taking 40 mg simvastatin (per C allele RR 2.6; 95% CI 1.3–5.0; 21 cases and 16,643 controls) [22]. Four other studies have confirmed this association between *SLCO1B1* and SRM with simvastatin [23,24,25,26].

The association of SRM with the *SLCO1B1* c.521T>C genotype has not been established for atorvastatin [23,24,26,27,28]. Only Puccetti et al. (2010) have reported a statistically significantly increased risk in 46 patients with familial hypercholesterolemia taking 20–40 mg atorvastatin per day [29]. When multiple statins were combined a statistically significant increased risk was found in five studies [23,24,25,26,30]. However, Linde et al. (2010) and Brunham et al. (2012) did not report a statistically significant increased risk in a case-control study of, respectively, 27 and 25 cases and 19 and 83 controls [27,31]. These two studies are in line with the results of Hubáček et al. (2015) in a Czech population of 3294 patients receiving 10–20 mg simvastatin or atorvastatin per day, who also did not find a statistically significant association [7].

The clinical relevance of PGx for statins remains debated because of the small effect sizes of other studies than the SEARCH Collaborative Group and number of patients that are expected to benefit from them, while the prevalence of the heterozygous *SLCO1B1* c.521T>C genotype in the general population is estimated to be 14–22% [15,22,32]. The prevalence of the relevant PGx variant, together with the large group of patients receiving statin prescriptions, might make even slight risk reductions relevant for practice. To be able to evaluate the contribution PGx delivers to effective and safe drug therapy, information on clinical validity and utility is needed. Clinical validity and utility of genetic tests should be assessed before implementation [33]. Clinical validity focuses on the discriminative ability of a test [33]. Clinical utility implies that a genetic test should impact on health outcomes in a relevant way in clinical practice [34,35,36]. Therefore, the net benefit of health outcomes also depends on the clinical context, such as when and where PGx is applied (Figure 1) [35,37]. PGx has already proven clinically useful in secondary health care [38,39]. However, as the CPIC and DPWG guidelines illustrate, PGx could also contribute to treatments in primary health care, such as statins [12,40].

By identifying the SLCO1B1 c.521T>C SNP pre-emptively or as companion diagnostics (CDx) (Figure 1), statin therapy can become more efficacious while SRM is prevented. However, the clinical validity and utility have not been established. The starting point of clinical validity is whether the SLCO1B1 risk genotype is associated with any ADR. Therefore, we aimed to analyze if the SLCO1B1 c.521T>C is associated with any change in statin therapy in participants in a population-based Dutch cohort study.

## 2. Materials and Methods

### 2.1. Study Population

The Doetinchem Cohort Study (DCS) is an ongoing longitudinal population-based cohort of randomly selected Dutch inhabitants of Doetinchem, aged 20–59 years at baseline (1987–1991) [41,42]. Participants have been re-examined once every five years since the baseline measurements. Data from the Doetinchem cohort were linked to the Out-patient Pharmacy Database of the PHARMO Database Network. This database comprises general practitioner- (GP) or specialist-prescribed healthcare products dispensed by the out-patient pharmacy. The dispensing records include information on type of product, date, strength, dosage regimen, quantity, route of administration, prescriber specialty, and costs. Informed consent for linkage between data of the Doetinchem cohort and the PHARMO Database Network was acquired before linkage was conducted [42].

Participants were included in this study when a blood sample was available and when they had at least one dispensing of simvastatin or atorvastatin between 1 July 1998 and 31 August 2015. While data were available from 1998 onwards, the starting date for our study (1 July 1998) was chosen to increase reliability on first start of statin therapy, because it was unknown whether patients were using statins before 1 January 1998. Patients with a dispensing before 1 July 1998 were excluded from the study population.

### 2.2. Follow-Up

After inclusion, participants were followed from the date of the first statin dispensing until (1) a follow-up period of three years was achieved, (2) the final date for the current analyses was reached (31 August 2015) within three years after follow-up, (3) the patient died, or (4) a loss to follow-up occurred. A period of three years of follow-up was chosen because the risk for SRM is the highest during the early use of statins [24].

### 2.3. Genotyping

Blood samples were taken during every (re-)examination and cryopreserved. DNA was extracted from buffy coats by a salting out method [43]. Using the QuantStudio™ 12 K Flex Real-time PCR system (Life Technologies, Bleiswijk, The Netherlands) with the PGx Express Panel array (ThermoFischer Scientific, Waltam, MA, USA), DNA was analyzed for the SNP c.521T>C in *SLCO1B1* according to the manufacturer’s protocol between September–November 2017.

### 2.4. Outcomes

We used proxies to assess the dispensing policy and the number of SRM. The first outcome was “the difference in dose change”. This was measured by comparing the dose category of the first dispensing to dose category of the last dispensing. The dose categories of simvastatin (ATC code C10AA01) were 10, 20 and 40 mg and atorvastatin (ATC code C10AA05) were 5, 10, 20, 40, and 80 mg based on the defined daily dose equivalent (DDD-E).

The second outcome was *any change in drug use.* This was categorized by discontinuation and switching. Discontinuation was defined as no new statins dispensed within 90 days after the theoretical end date of the last dispensing and no other cholesterol-lowering drug within 90 days after the end of the last dispensing or no further dispensing issued for any cholesterol-lowering drug and more than 90 days available to the right censoring date. Switching was defined as no new statins dispensed within 90 days after the theoretical end date of the last dispensing and another cholesterol-lowering drug within 90 days after the theoretical end date of the last dispensing [44,45]. Switching was specified by changing from simvastatin or atorvastatin to another statin (simvastatin, atorvastatin, rosuvastatin, pravastatin, or fluvastatin) or cholesterol-lowering drug (acipimox, ezetimibe, bezafibrate, ciprofibrate, fenofibrate, gemfibrozil, colesevelam, cholestyramine, nicotinic acid, docosapentaenoic acid, or eicosapentaenoic acid) [46].

The third outcome was *the difference in time to establish stable dosing*, defined as the time until three times successively the same dose was dispensed based on DDD-E. After this moment, the dosing regimen was assumed to be stable. The time to establish stable dosing was used as a measure for health impact, i.e., when a patient has a stable dosing regimen sooner, they are assumed to have less SRM and need less health care visits.

### 2.5. Covariables

#### 2.5.1. Potential Confounders

Potential confounders were identified before analysis and grouped into three categories: lifestyle factors (cigarette smoking, physical activity, BMI), biological factors (age, sex, diabetes mellitus, systolic blood pressure, HDL level), and drug-related factors (starting dose, concomitant drug use) [9,12,47,48]. Smoking was included because smokers are hypothesized to have a more sedentary lifestyle than non-smokers. As a result, they might have a lower risk of SRM. Specific concomitant drugs assessed were: amiodarone, diltiazem, verapamil, fluconazole, ketoconazole, clarithromycin, erythromycin, cyclosporine, gemfibrozil, and HIV protease inhibitors, as all inhibit statin metabolism through CYP3A4 or OATP1B1 proteins. The use of these drugs was analyzed as a yes/no variable. We had no data available on relevant co-morbidities.

#### 2.5.2. Effect Modifier

The type of initial statin was assumed an effect modifier, based on the different pathways in which simvastatin and atorvastatin are metabolized in the body. Therefore, groups were stratified according to the statin at first dispensing.

### 2.6. Statistical Analysis

The Hardy–Weinberg equilibrium was calculated to confirm that the allele frequency in the study population was constant and in balance by using a Chi-squared test. The genetic model was co-dominant, where each mutant allele contributes to the amount that a patient is affected, in line with the OR per C-allele reported by Link et al. (2008) [22]. Sample size for a power of 80% was calculated (Appendix A). Based on stratification for one variable and a dropout rate of 10%, sample size had to be at least 176 patients, approximately 44 per compared group. The outcome of the study used for the power calculation was myopathy and the daily dose was 80 mg of simvastatin. We expect that our sample needs to be bigger to correct for the regular lower dose and the less directly measured outcomes.

Baseline characteristics of the different genotype groups were compared by linear regression for continuous variables and Chi-squared tests for categorical variables to notice significant deviations. *The difference in time to establish the stable dosing* was analyzed with Cox proportional hazard analysis. *The difference in dose change* and *any change in drug use* (discontinuation and switch) were analyzed with logistic regression. Stratification was limited to initial statin, because stratification into smaller subgroups could result in not having enough power. All analyses were conducted with SPSS software (version 24.0). A *p*-value ≤ 0.05 was considered to be statistically significant. ORs, HRs, and test characteristics were calculated separately for simvastatin and atorvastatin.

## 3. Results

### 3.1. Population Sample

After applying the inclusion criteria, 2226 statin users could be linked between the PHARMO Database Network and 5683 DCS-participants (39.2%). For the current study, measurements for DCS participants were considered eligible if a participant had participated in at least three rounds of measurements. Therefore, another 1067 participants were excluded, resulting in 1159 included statin users (52.1%). Of the included statin users, 1136 (98.0%) were successfully genotyped for *SLCO1B1*. In Table 1, the baseline characteristics of our study population are shown. At baseline, 928 (81.7%) participants were using simvastatin and 208 (18.3%) atorvastatin. Start dose of simvastatin was lower in women compared to men (DDD eq. at baseline median 0.7 versus 1.3; *p* = 0.054). The genotypes were in Hardy–Weinberg equilibrium (χ^2^ = 0.009; *p* = 0.924) (Appendix A).

### 3.2. Dose Change

Among the simvastatin users, 31 (3.3%) participants changed to a lower dose at the end of follow-up compared to their start dose, and 8 (3.8%) patients in the atorvastatin group. There were no statistically significant differences between the TT and TC/CC genotype groups (Table 2).

### 3.3. Any Change in Drug Use: Discontinuation and Switching

Within the three years of follow-up, 535 (47.1%) of the 1136 statin users discontinued their treatment and 217 (19.1%) switched treatment. The discontinuation proportion between simvastatin and atorvastatin users was comparable (47.2% vs. 46.6%, *p* = 0.812), while more atorvastatin users switched their treatment compared to simvastatin users (27.4% vs. 17.2%, *p* = 0.001). Among simvastatin users, a TC/CC genotype was not significantly associated with any change in statin treatment (OR = 0.93, 95% CI 0.69–1.25) compared to a TT genotype. Users of atorvastatin with a TC/CC genotype did not change their treatment more often compared to patient with a TT genotype either (OR = 0.58, 95% CI 0.31–1.09) (Table 2).

### 3.4. Test Characteristics for SRM

There was no association between *SLCO1B1* 521T>C and switch or discontinuation, so the test characteristics of genotyping for *SLCO1B1* 521T>C were not calculated: sensitivity, specificity, positive predictive value, negative predictive value.

### 3.5. Time to Establish Stable Dosing

The median time to establish stable dosing was 90 days (IQR 97 days). Among the *SLCO1B1* TT genotype, the median time was 90 days (IQR 97 days), and among *SLCO1B1* TC/CC genotypes, 89 days (IQR 108). For simvastatin and atorvastatin users, it took a median number of 89 days (IQR 96 days) and 92 days (IQR 108 days), respectively, until successively three times the same dosing was dispensed. The time to establishing a stable dose was not significantly different between TT genotype and TC/CC genotypes, both for simvastatin users and for atorvastatin users (Table 2).

## 4. Discussion

In this study, we aimed to investigate the clinical validity and clinical utility of pre-emptive pharmacogenomics *SLCO1B1* c.521T>C screening in primary care. The starting point of clinical validity is whether the *SLCO1B1* risk genotype is associated with any change in statin therapy. In our cohort study, 30.5% of the participants had an actionable genotype (*n* = 316 TC, 27.8%, and *n* = 31 CC, 2.7%). The number of participants that discontinued treatment within 3 years (47.1%) was in line with previous findings [10,11]. We did not find a statistically significant association between *SLCO1B1* and any change in simvastatin or atorvastatin use. The time to stable dose was comparable between included participants regardless of genotype or statin used. In a randomized controlled trial comparing outcomes between usual care and PGx-informed prescription, Peyser et al. (2018) also did not find a difference in self-reported statin adherence between *SLCO1B1* groups [49]. We did find a statistically significant difference in the starting dose between males and females for simvastatin.

The DCS is a population-based cohort with a relatively long follow-up, enabling both cross-sectional and longitudinal comparisons [50], which increases its external validity. Furthermore, we have used routinely collected data of drug dispensing instead of prescriptions through community pharmacies, corrected for known confounders and stratified for type of statin as an effect modifier, and we applied multivariate methods to maximize internal validity. However, not all potential covariates could be analyzed, such as thyroid disorder and rheumatoid arthritis, because these data were not available for the DCS [9,48]. A major limitation is that we have used proxies to assess the frequency of SRM. As a result, we do not know if the dose changes and statin changes were actually due to SRM. Furthermore, the study may have been underpowered, despite the available sample size. The power calculation was based on the results from Link et al. [22], but in our cohort, 80 mg simvastatin was not used as a start dose. The sample size is, however, comparable to an average Dutch GP practice, and thus, gives a sufficient estimate of the effect for such a primary care practice. Finally, we used a framework for clinical validity and utility, and strived to analyze data beyond ORs which would provide important insights for statistically significant genotype associations [33].

To assess clinical validity and utility of *SLCO1B1*, we suggest a prospective study to monitor actual SRM instead of using proxies, including relevant lifestyle, drug-related, and biological factors [51,52]. SNPs in other genes alone or in combination with *SLCO1B1* c.521T>C might also influence the risk for SRM, such as the SNP c.421C>A in the *ABCG2* gene and 15389C>T on the CYP3A4 gene [5]. Additional insights might be gathered from an opportunistic genomic screening (OGS) approach [53]. OGS is defined as a deliberate search for genetic variants unrelated to the diagnostic question. Due to more and more people having their genomes sequenced, the potential to use virtual panels to actively look for additional variants increases [53]. PGx is not commonly included in OGS panels, while these might present a good option to collect real-world data. As with OGS-panels, the test context in which testing for *SLCO1B1* will be provided in practice might not be a single-SNP-based approach, but rather a panel-based approach [54]. Therefore, the validity and utility of such an approach will need to be established, including factors such as timing of testing and cost-effectiveness [55]. To study these aspects, outcomes such as the number needed to genotype and the population-attributable fraction of SRM should be studied. A current lack in literature is the reporting on clinical validity and utility [33]. When a study is designed and reported on, it is important to include relevant outcomes. Stakeholders should be included in the process to ensure that outcomes relevant to them are gathered and reported from research [56,57]. These include pharmacists, GPs, patients, researchers, clinical chemists, health care insurers, and policy makers.

## 5. Conclusions

Our study indicates that *SLCO1B1* c.521T>C screening is unlikely to impact relevant factors considering statin use, such as reaching a stable dose sooner and decreasing discontinuation and switches. It is unlikely that participants would have had a shorter time to establish stable dosing and would have experienced less SRM. Our findings do suggest that an appropriate method for treatment adherence should be studied, since approximately 47% patients discontinued their treatment, as well as follow-up or review of guidelines for initial statin treatment, because women had a lower starting dose than men. Furthermore, prospective studies focused on ADR need to take into account outcomes for clinical utility, such as predictive characteristics of (a panel of) SNPs and other outcomes relevant to medical doctors, patients, and other stakeholders.

## Figures and Tables

**Figure 1 genes-14-00456-f001:**
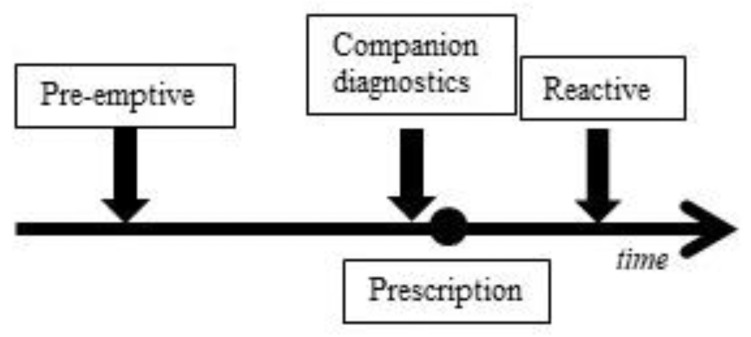
Pre-emptive testing is unrelated to a specific treatment and performed independent of a medical indication. Companion diagnostic testing is performed at the time of prescribing to choose the right drug or dosage. Reactive testing occurs after a patient has started a drug treatment. The aim of reactive testing is to find an explanation for side effects and improve an existing drug therapy.

**Table 1 genes-14-00456-t001:** Baseline characteristics of 1136 statin users from the Doetinchem cohort, categorized by the *SLCO1B1* c.521T>C genotype.

*SLCO1B1* Genotype	TT (*n* = 789)	TC (*n* = 316)	CC (*n* = 31)	Total
Sex				
Male, %	51.1	51.3	38.7	50.8
Female, %	48.9	48.7	61.3	49.2
Age (years), mean (±SD)	62.5 (9.1)	62.8 (9.1)	61.8 (9.4)	62.6 (9.1)
BMI (kg/m^2^), mean (±SD)	27.8 (4.5)	27.4 (4.5)	28.3 (4.4)	27.7 (4.6)
Diastolic blood pressure (mmHg), mean (±SD)	83.5 (10.9)	83.6 (10.4)	82.5 (11.0)	83.5 (10.7)
Systolic blood pressure (mmHg), mean (±SD)	136.9 (18.4)	136.0 (18.0)	136.5 (21.0)	136.5 (18.4)
Total cholesterol (mmol/L), mean (±SD)	5.9 (1.2)	5.9 (1.2)	6.2 (1.3)	5.9 (1.2)
Smoking				
Current, %	27.9	25.2	32.3	27.3
*Past*, %	45.9	45.5	48.4	45.8
*Never*, %	26.2	29.3	19.4	26.9
Statin dispensed				
*Simvastatin*, %	83.8	76.6	80.6	81.7
*Atorvastatin*, %	16.2	23.4	19.4	18.3
Start dose simvastatin (mg)				
*10* mg, %	11.0	8.8	12.0	10.4
*20* mg, %	37.7	45.8	52.0	40.2
*40* mg, %	51.3	45.4	36.0	49.3
Start dose atorvastatin (mg)				
*5* mg, %	0.3	0.0	0.0	0.2
*10* mg, %	57.0	54.1	66.7	56.3
*20* mg, %	33.6	36.5	33.3	34.6
*40* mg, %	6.3	8.1	0.0	6.7
*80* mg, %	1.6	1.4	0.0	1.4

**Table 2 genes-14-00456-t002:** The association between *SLCO1B1* c.521T>C genotype and discontinuation, switch, and time to establish a stable dose, by statin type.

		Simvastatin		Atorvastatin	
Dose Change	Event (%)	Crude OR (95% CI)	Adjusted OR ^a^ (95% CI)	Crude OR (95% CI)	Adjusted OR ^b^ (95% CI)
TT	30 (4.1)				
TC/CC	9 (3.0)	0.50 (0.19–1.31)	0.42 (0.15–1.16)	1.72 (0.42–7.11)	1.97 (0.43–9.04)
Total	39 (3.4)				
**Change (discontinuation or switch)**	**Event (%)**	**Crude OR (95% CI)**	**Adjusted OR ^a^ (95% CI)**	**Crude OR (95% CI)**	**Adjusted OR ^a^ (95% CI)**
TT	529 (67.1)				
TC/CC	223 (64.3)	0.93 (0.69–1.25)	0.80 (0.58–1.11)	0.58 (0.31–1.09)	0.57 (0.27–1.18)
Total	752 (66.2)				
**Time to establish stable dosing regimen**	**Median (IQR), days**	**Crude OR (95% CI)**	**Adjusted OR ^a^ (95% CI)**	**Crude OR (95% CI)**	**Adjusted OR ^a^ (95% CI)**
TT	90 (97)				
TC/CC	89 (108)	1.02 (0.87–1.21)	1.06 (0.89–1.26)	0.84 (0.61–1.14)	0.82 (0.57–1.16)
Total	90 (97)				

OR = Odds Ratio; 95% CI = 95% confidence interval. Adjusted for: ^a^. Age, Sex, Diabetes Mellitus, Systolic blood pressure, HDL level, BMI, cigarette smoking, physical activity, starting dose, concomitant drug use; ^b^. because of small sample size, a limited number of confounders could be added in the model: Age, Sex, Diabetes Mellitus, Systolic blood pressure, HDL level.

## Data Availability

The datasets generated during and/or analyzed during the current study are not publicly available, as formulated in the informed consent for participants, but are available from the corresponding author on reasonable request.

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
