# Peer review of "Predictive Value of SLCO1B1 c.521T>C Polymorphism on Observed Changes in the Treatment of 1136 Statin-Users"

_genes, 2023, doi:10.3390/genes14020456_

Round 1

Reviewer 1 Report

Overall, this is an important negative study which is valuable to publish.

In line 118, please define the acronym "GP."

In line 133, the sentence reads "A period of three years of follow-up was chosen because the risk of SRM is the highest during the first years of statin therapy."  It seems like there is a number missing between "first" and "years". I'm wondering if the authors meant to say "first three years" or "first year". Please clarify.

In line 169, cigarette smoking is mentioned as a potential confounder accounted for in the study design, but is not mentioned in the introduction as to what kind of effect it has. Please add a sentence in the introduction about the potential effect of cigarette smoking on statin adverse effects. 

In line 264, the authors state that the study has sufficient power through the available sample size. However, the power calculation only is based on simvastatin 80mg dose as stated in line 187.  I didn't see the simvastatin 80mg dose mentioned anywhere in the paper, nor the number of patients who were taking the 80mg dose during the three year follow up time. Can the authors provide a power calculation for the simvastatin 40mg dose? If not, then it should be stated as a limitation that the study may have been underpowered. 

Could the authors easily perform a post-hoc analysis to analyze the LDL-lowering efficacy amongst the different SLCO1B1 phenotypes? This would help strengthen the study given the pharmacokinetic differences expected between SLCO1b1 phenotypes.

In line 284-287,  recommend changing "would not" to "is unlikely to". It doesn't seem clear to me that "would not" helps communicate the negative results of the study.

I would like to thank the authors for their contribution. 

Author Response

Overall, this is an important negative study which is valuable to publish.

Thank you for your review and suggestions. We have incorporated them in the new version of the manuscript. You can find the point by point response below.

  • In line 118, please define the acronym "GP."

We have added “ general practitioner” in line 118.

  • In line 133, the sentence reads "A period of three years of follow-up was chosen because the risk of SRM is the highest during the first years of statin therapy."  It seems like there is a number missing between "first" and "years". I'm wondering if the authors meant to say "first three years" or "first year". Please clarify.

We have added “early use” in line 133. We indeed meant during the early use of statin treatment, referring back to the decision to use a follow-up time of three years.

  • In line 169, cigarette smoking is mentioned as a potential confounder accounted for in the study design, but is not mentioned in the introduction as to what kind of effect it has. Please add a sentence in the introduction about the potential effect of cigarette smoking on statin adverse effects.

We have added in the methods “Smoking was included because smokers are hypothesized to have a more sedentary life-style than non-smokers. As a result, they might have a lower risk of SRM.” The introduction is already quite lengthy, so therefore we added it in the methods.

  • In line 264, the authors state that the study has sufficient power through the available sample size. However, the power calculation only is based on simvastatin 80mg dose as stated in line 187.  I didn't see the simvastatin 80mg dose mentioned anywhere in the paper, nor the number of patients who were taking the 80mg dose during the three year follow up time. Can the authors provide a power calculation for the simvastatin 40mg dose? If not, then it should be stated as a limitation that the study may have been underpowered. 

We added the power calculation as a limitation in the discussion in line 265.

  • Could the authors easily perform a post-hoc analysis to analyze the LDL-lowering efficacy amongst the different SLCO1B1 phenotypes? This would help strengthen the study given the pharmacokinetic differences expected between SLCO1b1 phenotypes.

We unfortunately don’t have the data available to perform this analysis.

  • In line 284-287,  recommend changing "would not" to "is unlikely to". It doesn't seem clear to me that "would not" helps communicate the negative results of the study.

We have replaced “would not”.

I would like to thank the authors for their contribution. 

Reviewer 2 Report

The authors have carried out a case-control substudy within a larger cohort health study with genotyping for the SLCOB1 T-C mutation which has previously been studied in the same context with statins for ADR with doubtful significance. The present study is well-designed with appropriate power calculations so that the negative findings seem reliable and so contribute to clarification of this question, which is of importance taking the high prevalence of statin medication into consideration. The authors, who have used proxy measures for possible ADR have various relevant suggestions on how to further follow up on the results.

Author Response

Thank you for your compliments and review. We indeed think it is important to study more direct measurements of ADR, to get an even better estimation of the clinical validity and utility.

Reviewer 3 Report

Overall the manuscript is well written and easy to read. Main goal of study is well described as well as inclusion and exclusion criteria. However, no new information is provided related to previous publications in the field. It reinforces the absence of prediction for serious adverse side effects among statin users, particularly those carrying the specific polymorphism. Main relevance resides in analyzing a national cohort where adequate follow up is achieved. 

Several issues arose after a deeper review.

1) In the introduction the authors state that discontinuation is mainly related to side effects. However that it is not the case in many other countries. Many times is related to cost, access and adherence to long term therapy. What about other interventions such as diet, exercise and lifestyle changes? 

2) No information is provided related to reasons to increase or reduce statin dose. Was really based on side effects or simply a consequence of no effect in cholesterol levels?. No information is also provided in relation to reasons for switching among statins. 

3) The authors should consider beyond drug interaction, the dosing of those interactive drugs. In addition, co-morbidities that could influence switching among therapies. More information should be provided related to confounder analyses. 

4) What about prevalence of non alcoholic liver hepatic disease in this cohort? 

5) What about alcohol consume among users? 

6) What about thyroid disorders as common co-morbidity among women?

7) Based on the results, suggesting the preemptive search for this polymorphism when prescribing statins should be advised or not? is it cost-effective considering the low prevalence of severe side effects?. The authors should elaborate more in terms of recommendations. Suggesting a prospective trial to assess the relevance of this polymorphism should be linked to defined higher risk population. Based on current evidence, what potential users could be candidate for such a screening? The authors should discuss this aspect.

Author Response

Overall the manuscript is well written and easy to read. Main goal of study is well described as well as inclusion and exclusion criteria. However, no new information is provided related to previous publications in the field. It reinforces the absence of prediction for serious adverse side effects among statin users, particularly those carrying the specific polymorphism. Main relevance resides in analyzing a national cohort where adequate follow up is achieved. 

Thank you for your compliments and review. This study was designed to give real world insight in the potential clinical utility of prediction of SLCO1B1 genotype for prevention of side effects. This was done by retrospectively evaluating impact of (at that time unknown) genotype on statin use. Since predictive testing of SLCO1B1 for personalizing treatment with statins is highly debated in literature evidence of need for implementation is awaited. We feel the fact that this study does not show a significant effect, makes this study of added value to the existing evidence and in the debate. We have carefully considered your suggestions. You can find the point by point response below.

Several issues arose after a deeper review.

  • In the introduction the authors state that discontinuation is mainly related to side effects. However that it is not the case in many other countries. Many times is related to cost, access and adherence to long term therapy. What about other interventions such as diet, exercise and lifestyle changes? 

The referenced studies are international studies, including a systematic review. We therefore consider this statement supported by international data and not only the Dutch situation. Furthermore, we do not expect these factors to differ between the groups with different SLCO1B1 genotypes

  • No information is provided related to reasons to increase or reduce statin dose. Was really based on side effects or simply a consequence of no effect in cholesterol levels?. No information is also provided in relation to reasons for switching among statins. 

This is indeed one of the major limitations of our study, which we have addressed in the discussion. “Our major limitation is that we have used proxies to assess the frequency of SRM.” We have now elaborated on this limitation in the discussion: “As a result, we do not know if the dose-changes and statin changes were actually due to SRM.”

  • The authors should consider beyond drug interaction, the dosing of those interactive drugs. In addition, co-morbidities that could influence switching among therapies. More information should be provided related to confounder analyses. 

We have added in line 176 “The use of these drugs was analyzed as a yes/no variable. We had no data available on relevant co-morbidities.” In the discussion we also point this out in line 264 “However, not all potential covariates could be analyzed, such as thyroid disorder and rheumatoid arthritis, because this data was not available for the DCS.

  • What about prevalence of non alcoholic liver hepatic disease in this cohort? 

This data unfortunately was not available.

  • What about alcohol consume among users? 

This data unfortunately was not available.

  • What about thyroid disorders as common co-morbidity among women?

This data unfortunately was not available.

  • Based on the results, suggesting the preemptive search for this polymorphism when prescribing statins should be advised or not? is it cost-effective considering the low prevalence of severe side effects?. The authors should elaborate more in terms of recommendations. Suggesting a prospective trial to assess the relevance of this polymorphism should be linked to defined higher risk population. Based on current evidence, what potential users could be candidate for such a screening? The authors should discuss this aspect.

Based on our results we cannot recommend a pre-emptive search for only this polymorphism. We hope to make clear in our conclusion that this study suggests little clinical influence of SLCO1B1 genotype on statin use. Because of the limitations of the study it is however difficult to state that there is absolutely no added value. See line 306: “Our study indicates that SLCO1B1 c.521T>C screening is unlikely to impact relevant factors considering statin use, such as reaching a stable dose sooner and decreasing dis-continuation and switches. It is unlikely that participants would have had a shorter time to establish stable dosing and would have experienced less SRM.” Therefore we advise to study a panel with different PGx-variants. The population to include in such a study can be a general population, because it will test for a range of PGx-variants, and therefore different risk groups.

Round 2

Reviewer 3 Report

The authors have properly answered my observations